# Low-Dose Rifabutin Increases Cytotoxicity in Antimitotic-Drug-Treated Resistant Cancer Cells by Exhibiting Strong P-gp-Inhibitory Activity

**DOI:** 10.3390/ijms23137383

**Published:** 2022-07-02

**Authors:** Ji Sun Lee, Yunmoon Oh, Hyung Sik Kim, Sungpil Yoon

**Affiliations:** School of Pharmacy, Sungkyunkwan University, 2066 Seobu-ro, Jangan-gu, Suwon 16419, Korea; leejs7186@naver.com (J.S.L.); oym9083@g.skku.edu (Y.O.); hkims@skku.edu (H.S.K.)

**Keywords:** antibiotics, rifabutin, co-treatment, cancer, P-gp, drug resistance

## Abstract

The cytotoxicity of various antibiotics at low doses in drug-resistant cancer cells was evaluated. Low doses of rifabutin were found to markedly increase the cytotoxicity of various antimitotic drugs, such as vincristine (VIC), to P-glycoprotein (P-gp)-overexpressing antimitotic-drug-resistant KBV20C cells. Rifabutin was also found to exert high levels of P-gp-inhibitory activity at 4 and 24 h posttreatment, suggesting that the cytotoxicity of VIC + rifabutin was mainly due to the direct binding of rifabutin to P-gp and the reduction of VIC efflux by P-gp. The combination of VIC + rifabutin also increased early apoptosis, G2 arrest, and the DNA damaging marker, pH2AX protein. Interestingly, only the combination of VIC + rifabutin induced remarkable levels of cytotoxicity in resistant KBV20C cells, whereas other combinations (VIC + rifampin, VIC + rifapentine, and VIC + rifaximin) induced less cytotoxicity. Such finding suggests that rifabutin specifically increases the cytotoxicity of VIC in KBV20C cells, independent of the toxic effect of the ansamycin antibiotic. Only rifabutin had high P-gp-inhibitory activity, which suggests that its high P-gp-inhibitory activity led to the increased cytotoxicity of VIC + rifabutin. As rifabutin has long been used in the clinic, repositioning this drug for P-gp-overexpressing resistant cancer could increase the availability of treatments for patients with drug-resistant cancer.

## 1. Introduction

Antimitotic drugs, such as vincristine (VIC), eribulin, vinorelbine, vinblastine, paclitaxel, and docetaxel, can inhibit cellular division by preventing chromosomal segregations that occur via microtubule polymerization or depolymerization [1,2,3,4]. However, patients administered these drugs can develop multidrug resistance (MDR) [4,5,6]. Stem cells in solid tumors have been characterized for MDR cancer types [7]. Identifying novel drugs and their underlying mechanisms in MDR is important to facilitate the fast and effective treatment of cancer patients with MDR.

P-glycoprotein (P-gp) is located in the cellular membrane and is responsible for the efflux of anticancer drugs [8]. The over-expression of P-gp is a well-known mechanism underlying the MDR of cancer cells [8,9]. Although many inhibitors with P-gp activity have been evaluated in the clinic, their toxicities to normal tissues have prevented their use in treating patients with drug-resistant cancer [8,9,10,11]. Accordingly, researchers have dedicated their efforts to develop P-gp inhibitors with reduced toxicities to normal cells and specifically target drug-resistant cancer cells [8,9,10,11]. Previously, novel mechanisms and inhibitors that target P-gp activity with low toxicity to normal cells were recognized. Therefore, our aim was to reposition drugs known for their toxicity with Food and Drug Administration (FDA) approval [12,13]. Once instances of novel, FDA-approved drug repositioning are found for sensitizing P-gp-overexpressing resistant cancers, we believe that these drugs may be administered to patients with MDR cancer without the performance of further toxicity tests.

In this study, we examined novel repositioned drugs and their possible application for the treatment of P-gp-overexpressing resistant cancer cells, including the increased sensitization efficacy of repositioned drugs when used in combination with antimitotic drugs. Antibiotics (doxycycline, levofloxacin, fluoroquinolones, nitroxoline, rifabutin, etc.) exhibit anticancer activity and have been used in clinical trials as repositioned drugs for cancer patients [14,15,16,17]. Further, we evaluated antibiotics with potential cytotoxic effects when administered as a co-treatment with antimitotic drugs.

Rifabutin is an antibiotic used to treat tuberculosis. Rifabutin can inhibit DNA-dependent RNA polymerase in mycobacteria [18,19] and can be combined with anticancer drugs to eradicate *Helicobacter pylori* in stomach cancer patients to prevent relapse [18,19]. However, the potential use of rifabutin to target resistant cancer cells has not yet been explored. In this study, we investigated whether rifabutin, an FDA-approved drug, could increase the cytotoxicity of antimitotic drugs to P-gp-overexpressing MDR cancer cells. In addition, we determined whether rifabutin could inhibit P-gp activity in MDR cancer cells. Collectively, our results may facilitate the development of rifabutin-based therapies for drug-resistant cancers.

## 2. Results

### 2.1. VIC + Rifabutin Exhibits Increased Cytotoxicity in P-gp-Overexpressing Drug-Resistant KBV20C Cancer Cells

We aimed to identify novel repurposable drugs that could increase the cytotoxicity of antimitotic drugs to P-gp-overexpressing drug-resistant cancer cells. Various antibiotic drugs can be used in cancer therapy [14,15,17]. Thus, we attempted to identify novel repositioned drugs from antibiotics and determine whether the co-treatment of various antibiotics could increase the cytotoxicity of antimitotic drugs in drug-resistant KBV20C cancer cells.

As shown in Figure 1A, a single treatment with 10 μM rifabutin did not induce any cytotoxicity in P-gp-overexpressing KBV20C cells. However, co-treatment with 5 μM rifabutin and VIC markedly reduced the proliferation of VIC-treated KBV20C cells compared to single treatment with either VIC or rifabutin (Figure 1A,B). The highly cytotoxic effects of VIC + rifabutin on resistant KBV20C cells were confirmed using microscopic observations (Figure 1C). To determine the efficiency of rifabutin when combined with VIC in resistant KBV20C cells, 0.25, 0.5, 1, and 5 μM of rifabutin were tested. Based on our results, 1 μM rifabutin was sufficient to increase the cytotoxicity of VIC in VIC-treated resistant KBV20C cells (Figure 1C). Thus, a low dose of rifabutin is sufficient to increase the cytotoxicity of VIC when both drugs are co-administered to drug-resistant KBV20C cells. We proceeded to determine whether co-treatment with low-dose rifabutin and VIC could sensitize VIC-treated resistant KBV20C cancer cells and lead to long-term survival. Colony-forming assays were performed at 10 days after drug treatment. As shown in Figure 1D, 0.5 and 1 μM rifabutin reduced colony formation in VIC-treated KBV20C cells, whereas single treatment with either VIC or rifabutin led to similar colony size and number as compared with those of the control. Thus, the combination of VIC and rifabutin can be used to treat VIC-resistant cancer cells with long-term efficacy.

### 2.2. Low-Dose Rifabutin Has Strong P-gp-Inhibitory Activity after 4 h of Treatment

To investigate the mechanisms involved in the effect of VIC + rifabutin, the P-gp-inhibitory activity of rifabutin was evaluated in P-gp-overexpressing drug-resistant KBV20C cancer cells. P-gp inhibition by rifabutin was assumed to result in a high cytotoxic effect in VIC-treated KBV20C cells.

As aripiprazole, reserpine, and tariquidar display strong P-gp-inhibitory activities [20,21,22,23], they were used as positive controls. These well-known P-gp inhibitors were highly cytotoxic to P-gp-overexpressing KBV20C cells owing to their strong P-gp-inhibiting activity [20,21,22,23]. The number of KBV20C cells that accumulated rhodamine123 (a well-known P-gp substrate) following treatment with the potential P-gp inhibitors or positive controls was determined.

As shown in Figure 2A–C, approximately seven- to ninefold higher P-gp-inhibitory activity was obtained with the positive controls (aripiprazole, reserpine, and tariquidar) than with the DMSO-treated negative control. Further, rifabutin markedly increased P-gp-inhibitory activity, with a level similar to that obtained with the positive controls (Figure 2A–C). Such finding suggests that P-gp inhibition by rifabutin plays a primary role in the cytotoxic effect exhibited by VIC + rifabutin.

Rifabutin displayed similar strong P-gp-inhibitory activity at 4 h (short time) and 24 h (long time) after treatment (Figure 2A,B). Such a finding suggests that the sensitizing mechanism of rifabutin involves the inhibition of P-gp activity but not the repression of P-gp transcription or translation. Overall, low-dose rifabutin can increase the cytotoxicity of VIC in P-gp-overexpressing KBV20C cells by strongly inhibiting the pumping-out ability of P-gp.

### 2.3. Rifabutin Causes Dose-Dependent Increases in Cytotoxicity in VIC-Treated KBV20C Cells by Inducing Early Apoptosis

To further clarify the cytotoxic mechanism of action of VIC + rifabutin, apoptotic death was analyzed using annexin V staining. The distribution of apoptotic cells in either the early stage or the late stage was quantitatively estimated. As shown in Figure 2D, early apoptotic cells increased as the dose of rifabutin increased. In fact, 0.25, 0.5, 1, 2.5, and 5 μM rifabutin resulted in 4%, 8%, 10%, 13%, and 18% increases in early apoptotic cell death, respectively (Figure 2D). These results suggest that early apoptosis in VIC + rifabutin–treated cells can be markedly increased by increasing the concentration of rifabutin. However, an increase in late apoptotic death was not detected with increases in the dose of rifabutin (Figure 2D), suggesting that early apoptotic induction was induced by the cytotoxic effects of VIC + rifabutin.

The expression level of C-PARP, a well-known apoptotic marker, was determined to confirm increased apoptosis in VIC + rifabutin–treated cells at the molecular level. As shown in Figure 3A, C-PARP production was increased in VIC + rifabutin–treated cells. When C-PARP expression levels in cells treated with different concentrations of rifabutin (between 1 and 2.5 μM rifabutin) were compared, C-PARP expression in the VIC-treated resistant KBV20C cells was found to be increased by rifabutin in a dose-dependent manner, whereas its expression level did not markedly change after single treatment with either rifabutin or VIC.

### 2.4. Rifabutin Induces G2-Arrest and Increases DNA Damage in VIC-Treated Resistant KBV20C Cells

Fluorescence-activated cell sorting (FACS) analyses were performed to further determine whether cell cycle arrest was involved in the early apoptotic death of cells after co-treatment with VIC + rifabutin. The number of cells distributed in the G1, S, and G2 stages following drug treatment was quantified. As shown in Figure 3B, G2-arrested cells increased as the dose of rifabutin increased. In fact, 0.25, 0.5, 1, 2.5, and 5 μM rifabutin resulted in 24%, 27%, 35%, 54%, and 56% increases in G2-arrest cells, respectively (Figure 3B). These results suggest that G2 arrest after co-treatment with VIC + rifabutin can be increased by increasing the concentration of rifabutin.

To further investigate the expression of proteins involved in G2 arrest [2,4,21], Western blotting analysis was performed. As shown in Figure 3A, CDK protein expression after VIC + rifabutin treatment did not qualitatively differ from that after single treatment with either agent. Importantly, the expression level of pH2AX, a DNA damage marker, markedly increased in a dose-dependent manner following co-treatment with VIC + rifabutin (Figure 3A), suggesting that DNA damage might increase G2 arrest in VIC + rifabutin co-treated KBV20C cells. Thus, the DNA damage signal increases early apoptosis via G2 arrest in VIC + rifabutin co-treated KBV20C cells.

### 2.5. Co-Treatment with Rifabutin Increases the Cytotoxicity of Other Antimitotic Drugs in KBV20C Cells

We investigated whether rifabutin could increase the cytotoxicity of other antimitotic drugs by determining the cytotoxic effects of 5 μM rifabutin in combination with 0.1 μg/mL vinorelbine, 5 nM vinblastine, and 30 nM eribulin, which are antimitotic drugs used as chemotherapeutic agents for cancer patients [1,2,3]. As shown in Figure 4A, vinorelbine + rifabutin, vinblastine + rifabutin, and eribulin + rifabutin sensitized P-gp-overexpressing resistant KBV20C cells. The sensitizing effects of single treatments were similar to that of the control.

The cytotoxicity of VIC + rifabutin, vinorelbine + rifabutin, vinblastine + rifabutin, and eribulin + rifabutin was quantitatively analyzed using annexin V analysis. As shown in Figure 4B, vinorelbine + rifabutin, vinblastine + rifabutin, and eribulin + rifabutin caused remarkable increases in early apoptosis, similar to VIC + rifabutin. These results confirm that vinorelbine + rifabutin, vinblastine + rifabutin, and eribulin + rifabutin had a similar cytotoxic effect to VIC + rifabutin in sensitizing drug-resistant KBV20C cancer cells. Thus, rifabutin could be combined with other antimitotic drugs to sensitize cancer cells overexpressing P-gp.

The sensitizing effect of a well-known P-gp inhibitor, verapamil, on VIC-treated KBV20C cells was compared to that of rifabutin. As shown in Figure 4C, rifabutin (2.5 μM) and verapamil (10 μM) produced similar sensitization effects in cells co-treated with VIC, suggesting that a low dose of rifabutin could increase the cytotoxicity in VIC-treated KBV20C cells to levels similar to that obtained with verapamil in sensitizing drug-resistant cancer cells overexpressing P-gp.

### 2.6. Other Antibiotic Drugs (Rifampin, Rifapentine, and Rifaximin) Exhibit Minor P-gp-Inhibitory Activity

As rifabutin is an antibiotic drug that is used to inhibit DNA-dependent RNA polymerase in mycobacterium, we determined whether P-gp-overexpressing resistant KBV20C cells could be sensitized by other ansamycin antibiotic drugs with DNA-dependent RNA polymerase inhibitory activities. Previous studies have reported that ansamycin antibiotic drugs (rifampin, rifapentine, and rifaximin) have RNA polymerase inhibiting activities [18,19,24]. Therefore, the cytotoxic effects of rifabutin were compared with those of antibiotics. Based on the microscopic observations and the degree of apoptosis induction shown in Figure 5A,B, 2.5 μM rifampin, rifapentine, and rifaximin exerted markedly lower cytotoxicity in VIC-co-treated KBV20C cells than 2.5 μM VIC + rifabutin. Rifabutin is assumed to have a specific structure for targeting P-gp-overexpressing resistant cancer, unlike other RNA polymerase targeting antibiotics (rifampin, rifapentine, or rifaximin). Thus, rifabutin can increase the cytotoxicity of VIC in P-gp-overexpressing drug-resistant KBV20C cells with toxic mechanisms independent of DNA-dependent RNA polymerase. Interestingly, when the P-gp-inhibitory activity of those antibiotics was measured, only rifabutin was found to have strong P-gp-inhibiting activity (Figure 5C). Such a finding suggests that the P-gp-inhibiting activity of rifabutin plays a key role in the cytotoxicity of VIC + rifabutin. In summary, among the four tested antibiotics with DNA-dependent RNA polymerase inhibitors, only rifabutin at a low dose was highly cytotoxic to VIC-treated resistant KBV20C cancer cells.

## 3. Discussion

Drug repositioning is a popular practice for clinically used drugs [12,13]. Drug repositioning is advantageous as it leads to wider clinical applications for cancer patients without the performance of additional toxicity tests. Previously, we investigated several repurposed drugs, such as antimalarial drugs, anti-HIV drugs, anti-allergic drugs, antipsychotic drugs, and tyrosine kinase inhibitors, to determine their effects on P-gp-overexpressing drug-resistant cancer cells [20,25,26,27].

The use of antibiotics in cancer therapy has been suggested. Since antibiotics have been used in the clinic for a long time, if they are found to display novel mechanisms for sensitizing resistant cancers, they can be easily applied to treat cancer patients. As a result, we opted to search for antibiotics that can increase the cytotoxicity of antimitotic drugs against P-gp-overexpressing drug-resistant cancers. To the best of our knowledge, this is the first study to demonstrate that co-treatment with rifabutin induced remarkably increased levels of cytotoxicity in VIC-treated P-gp-overexpressing resistant cancer cells. Herein, VIC + rifabutin displayed remarkably increased cytotoxicity in P-gp-overexpressing drug-resistant KBV20C cancer cells compared with VIC or rifabutin alone. Such a finding suggests that rifabutin is highly effective when administered with VIC in patients with resistant cancers and ineffective when administered as a monotherapy. As low-dose rifabutin could sensitize KBV20C cells treated with VIC, it may be useful in clinical settings owing to its minimal toxic concentration in normal cells. Our findings may facilitate the quick application of rifabutin in P-gp-overexpressing resistant patients as part of a combination therapy with antimitotic drugs. Our findings could also be applied to co-treatment with other types of antimitotic drugs. Herein, rifabutin exhibited similar sensitization effects in vinorelbine-, vinblastine-, and eribulin-treated resistant KBV20C cells. Accordingly, we hypothesized that rifabutin could be co-administered with various anticancer drugs to sensitize MDR cancer cells.

A low dose of rifabutin was found to have strong P-gp-inhibitory activity, indicating that the increased apoptosis induced by VIC + rifabutin was due to the P-gp-inhibitory activity of rifabutin, which prevented the efflux of VIC. Our findings indicate that rifabutin displayed similar efficacy to the established strong P-gp inhibitors, aripiprazole, reserpine, and tariquidar (positive controls), at comparably lower doses. As P-gp inhibitors exhibit toxicity in normal cells [20,21,22,23], rifabutin with strong P-gp-inhibitory activity could be considered as a co-treatment drug to specifically target P-gp-overexpressing resistant cancer cells. As personalized medicines are gaining popularity, our findings regarding rifabutin might contribute to effective prescriptive options for P-gp-overexpressing drug-resistant cancer patients. When verapamil and rifabutin were compared, rifabutin was assumed to induce nanomolar ranges of P-gp-inhibitory activity in an in vitro test-tube assay. In future studies, in vitro assays should be used to determine whether rifabutin directly binds to P-gp and inhibits its activity.

We sought to determine whether other antibiotics (rifampin, rifapentine, and rifaximin) used as DNA-dependent RNA polymerase inhibitors exhibited sensitization effects on VIC-treated KBV20C cells when administered at low doses. However, rifampin, rifapentine, and rifaximin did not display similar sensitization effects to rifabutin at low doses. Further, only rifabutin displayed strong P-gp-inhibitory activity, suggesting that the RNA polymerase inhibiting activity of rifabutin was not responsible for its ability to increase the cytotoxicity of VIC in P-gp-overexpressing resistant KBV20C cells. This finding indicates that the strong P-gp-inhibitory activity of rifabutin mainly contributes to its high cytotoxic effect on P-gp-overexpressing resistant KBV20C cancer cells treated with VIC.

The molecular mechanisms underlying the co-treatment of VIC + rifabutin were assessed to accelerate the application of rifabutin in the clinic, especially for patients with antimitotic-drug-resistant cancer. By performing a more detailed quantitative annexin V study, VIC + rifabutin was found to significantly increase early apoptosis of KBV20C cells. Based on microscopy, FACS, and annexin V analyses, early apoptosis was upregulated by treatment with rifabutin, which caused increased G2 arrest and reduced proliferation of P-gp-overexpressing resistant KBV20C cells. Furthermore, the expression of a DNA-damage-related marker, pH2AX, was markedly elevated, indicating that high DNA damage results in an increased number of cells undergoing early apoptotic death. Further in vivo studies using animal models are needed to facilitate the quick application of rifabutin for patients with MDR. The results obtained with the oral KBV20C cancer cell line may be applicable to other organ-generated P-gp-overexpressing resistant cancer cells.

In conclusion, rifabutin was found to markedly increase the cytotoxicity of VIC in P-gp-overexpressing drug-resistant KBV20C cells owing to its high P-gp-inhibitory activity. As rifabutin has long been used in the clinic, this drug could be repositioned for P-gp-overexpressing resistant cancer, enabling faster treatment of patients with drug-resistant cancer.

## 4. Methods and Materials

### 4.1. Reagents and Cell Culture

Rhodamine123 and verapamil were purchased from Sigma-Aldrich (St. Louis, MO, USA). VIC, vinorelbine, and vinblastine were purchased from Enzo Life Sciences (Farmingdale, NY, USA). Rifabutin, rifampin, rifapentine, and rifaximin were purchased from Selleckchem (Houston, TX, USA). Aqueous solutions of eribulin (Eisai Korea, Seoul, South Korea) were obtained from the National Cancer Center of South Korea.

Antibodies against C-PARP were obtained from Cell Signaling Technology (Danvers, MA, USA). Antibodies against α-LC3B and CDK1 were obtained from Abcam (Cambridge, UK). Antibodies against CDK2 and GAPDH were obtained from Santa Cruz Biotechnology (Santa Cruz, CA, USA). Antibodies against pH2AX were obtained from Sigma-Aldrich (St. Louis, MO, USA).

The human oral squamous carcinoma cell line, KB, and its multidrug-resistant subline, KBV20C, have been previously described [27]. All cell lines were cultured in RPMI 1640 containing 10% fetal bovine serum, 100 U/mL penicillin, and 100 μg/mL streptomycin (WelGENE, Daegu, South Korea).

### 4.2. Microscopic Observation

Cellular growth was observed with a microscope as previously described [27,28]. Briefly, cells were grown in 60-mm-diameter dishes and treated with drugs for 24 h. These cells were then examined immediately in two independent experiments using an ECLIPSETs2 inverted routine microscope (Nikon, Tokyo, Japan) with a ×40 or a ×100 objective lens.

### 4.3. Cell Viability Assay

Cell proliferation was measured by a colorimetric assay using an EZ-CyTox cell viability assay kit (Daeillab, South Korea) as previously described [20,27,28]. Briefly, cells were grown in 96-well plates and treated with drugs for 48 h. These cells were then incubated with 10 μL of EZ-CyTox solution at 37 °C for 1–2 h. Absorbance at 450 nm was measured immediately using a VERSA MAX Microplate Reader (Molecular Devices Corp., Sunnyvale, CA, USA). All experiments were performed at least in triplicate and repeated twice.

### 4.4. Colony Forming Assay

A colony forming assay was used to assess long-term growth in the presence of a drug based on a previously described method [22,23]. Briefly, 1–2 × 10^3^ cells were grown in 6-well plates for 5–6 days and then stimulated with 5 nM vincristine, 0.5 μM rifabutin, 1 μM rifabutin, 5 nM vincristine + 0.5 μM rifabutin, 5 nM vincristine + 1 μM rifabutin, or 0.1% DMSO (control) for 5–6 days. Fresh medium containing the drug(s) was changed twice. Colony forming assays were immediately performed using crystal violet staining after 10–12 days. Viable colonies were fixed with methanol, stained with 0.05% crystal violet for 20 min, washed with phosphate-buffered saline (PBS), and air-dried. Relative colonies were analyzed using an image analyzer. All experiments were repeated at least twice.

### 4.5. Fluorescence-Activated Cell Sorting (FACS) Analysis

FACS analysis was performed as previously described [20,27]. Cells were grown in 60-mm-diameter dishes and treated with drugs for 24 h. Cells were then dislodged by trypsin and pelleted by centrifugation. Cell pellets were washed thoroughly with PBS, suspended in 75% ethanol for at least 8 h at 4 °C, washed with PBS, and re-suspended in a cold propidium iodide (PI) staining solution (100 μg/mL RNase A and 50 μg/mL PI in PBS) for 30 min at 37 °C. These stained cells were analyzed in two independent experiments for relative DNA content using a Novocyte Flow cytometer (ACEA Biosciences, San Diego, CA, USA).

### 4.6. Annexin V Analysis

Annexin V analysis was conducted using an annexin V-fluorescein isothiocyanate (FITC) staining kit (BD Bioscience, Franklin, NJ, USA) as previously described [20,27]. Cells were grown in 60-mm-diameter dishes and treated with drugs for 24 h. These cells were then dislodged using trypsin and pelleted by centrifugation. The cell pellet was washed with PBS. After adding 5 μL of annexin V-FITC and 5 μL of PI to cells in 100 μL of binding buffer, the mixture was then incubated at room temperature for 30 min. Stained cells were analyzed in two independent experiments using a Novocyte Flow cytometer (ACEA Biosciences, San Diego, CA, USA).

### 4.7. Rhodamine123 Uptake Tests

To assess the ability of a drug to inhibit P-gp, rhodamine123 uptake tests were performed as described previously [20,27]. Briefly, cells grown in 60-mm-diameter dishes were treated with 10 μM verapamil, 2.5 μM aripiprazole, 5 μM reserpine, 1.5 μM tariquidar, or 1–5 μM rifabutin and then incubated at 37 °C for 1 h. After removing the medium, cells were washed with PBS. Cells stained for 3 h were analyzed in two independent experiments using a Novocyte Flow cytometer (ACEA Biosciences, San Diego, CA, USA).

### 4.8. Western Blot Analysis

Total cellular proteins were extracted as described previously [28,29]. Briefly, cells grown in 60 mm dishes were washed twice with cold PBS and detached. For total protein isolation, cells were suspended in a PRO-PREP™ protein extract solution (iNtRON, Seongnam, Korea) and placed on ice for 30 min. The suspension was collected after centrifugation at 15,000× *g* for 15 min at 4. Protein concentrations were measured using a protein assay kit (Bio-Rad, Hercules, CA, USA) according to the manufacturer’s instructions. Proteins were resolved by sodium dodecyl sulfate–polyacrylamide gel electrophoresis (SDS-PAGE) and subjected to Western blot analysis as previously described [28,29].

### 4.9. Statistical Analysis

All data are presented as mean ± S.D. from two independent experiments performed in triplicate. All statistical analyses were performed using one-way analysis of variance (ANOVA) followed by Bonferroni’s test. Analysis was performed using Graph Pad Prism Software Version 5.0 (GraphPad Software, San Diego, CA, USA). A *p*-value of less than 0.05 was considered statistically significant.

## Figures and Tables

**Figure 1 ijms-23-07383-f001:**
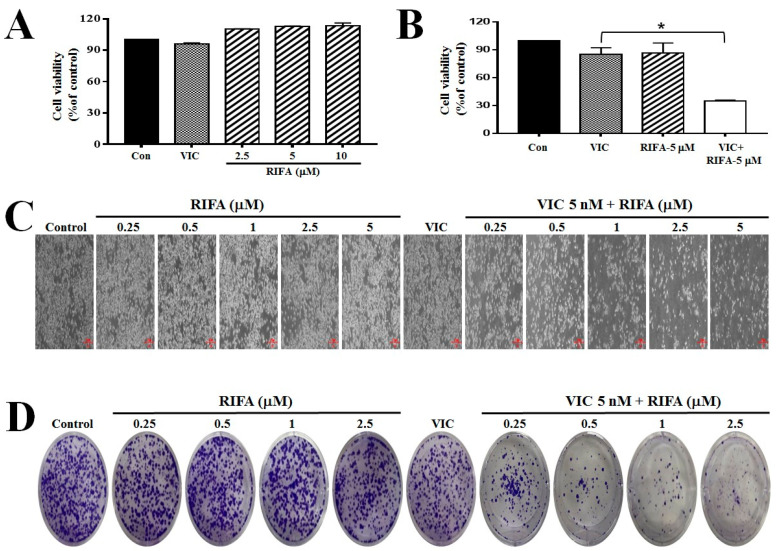
Rifabutin increases the cytotoxicity of VIC to P-gp-overexpressing drug-resistant KBV20C cancer cells. (**A**,**B**) KBV20C cells were treated with 5 nM VIC, indicated concentrations (μM) of rifabutin (RIFA), VIC + RIFA (5 μM), or 0.1% DMSO (CON) for 48 h. Cell viability assay was performed. Data are presented as mean ± SD. * *p* < 0.05 was considered be statistically significant. (**C**) KBV20C cells were treated with 5 nM VIC, the indicated concentration (μM) of rifabutin (RIFA) alone, or in combination with 5 nM VIC, or 0.1% DMSO (Control). After 1 day, all cells were observed using an inverted microscope at ×40 magnification. (**D**) KBV20C cells were treated for 3 days with 5 nM VIC, indicated concentration (μM) of rifabutin (RIFA) alone, or in combination with 5 nM VIC, or 0.1% DMSO (Control). Colony forming assays were immediately performed with crystal violet staining after a total of 10 days.

**Figure 2 ijms-23-07383-f002:**
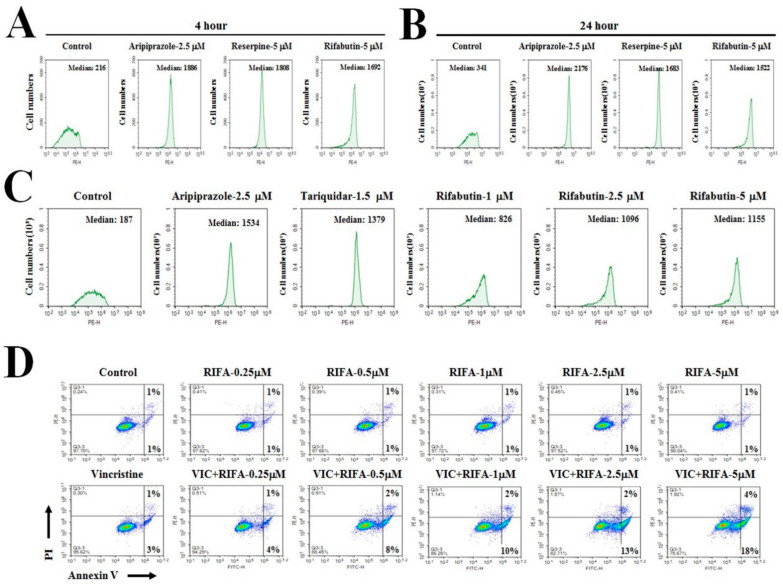
Rifabutin has high P-gp-inhibitory activity and increases early apoptosis in VIC-treated KBV20C cells. (**A**–**C**) KBV20C cells were treated with indicated concentrations of aripiprazole, reserpine, tariquidar, rifabutin (RIFA), or 0.1% DMSO (Control). After 4 h (**A**), 24 h (**B**), or 4 h (**C**) of rhodamine123 staining, all cells were examined using FACS analysis. (**D**) KBV20C cells were treated with 5 nM VIC, the indicated concentration (μM) of rifabutin (RIFA) alone, or in combination with 5 nM VIC, or 0.1% DMSO (Control). After 24 h, annexin V analyses were performed as described in Section 4: Materials and Methods.

**Figure 3 ijms-23-07383-f003:**
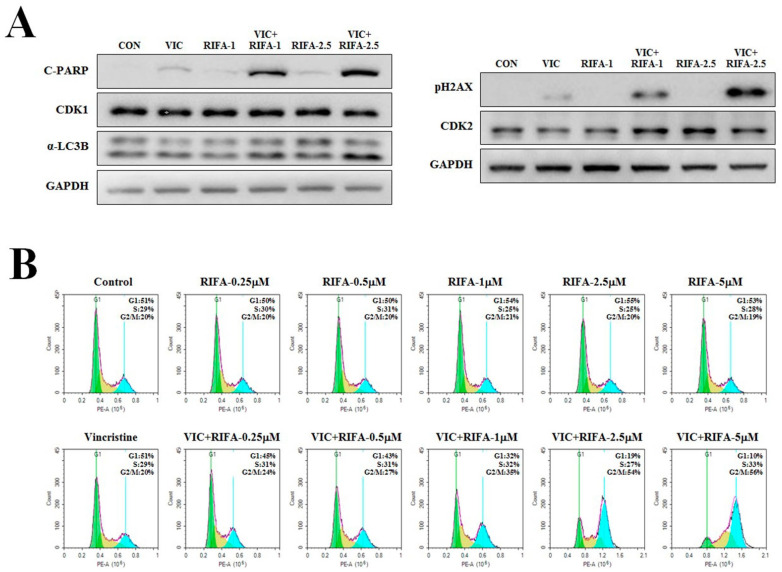
Rifabutin increases DNA damage and G2 arrest in VIC-treated KBV20C cells. (**A**) KBV20C cells were treated with 5 nM vincristine (VIC), 1 μM rifabutin (RIFA-1), 2.5 μM rifabutin (RIFA-2.5), 5 nM VIC with 1 μM rifabutin (VIC + RIFA-1), 5 nM VIC with 2.5 μM rifabutin (VIC + RIFA-2.5), or 0.1% DMSO (CON). After 24 h, Western blot analysis was performed using antibodies against C-PARP, CDK1, α-LC3B, pH2AX, CDK2, and GAPDH. (**B**) KBV20C cells were treated with 5 nM VIC, the indicated concentration (μM) of rifabutin (RIFA) alone, or in combination with 5 nM VIC, or 0.1% DMSO (Control). After 24 h, FACS analyses were performed as described in Section 4: Materials and Methods.

**Figure 4 ijms-23-07383-f004:**
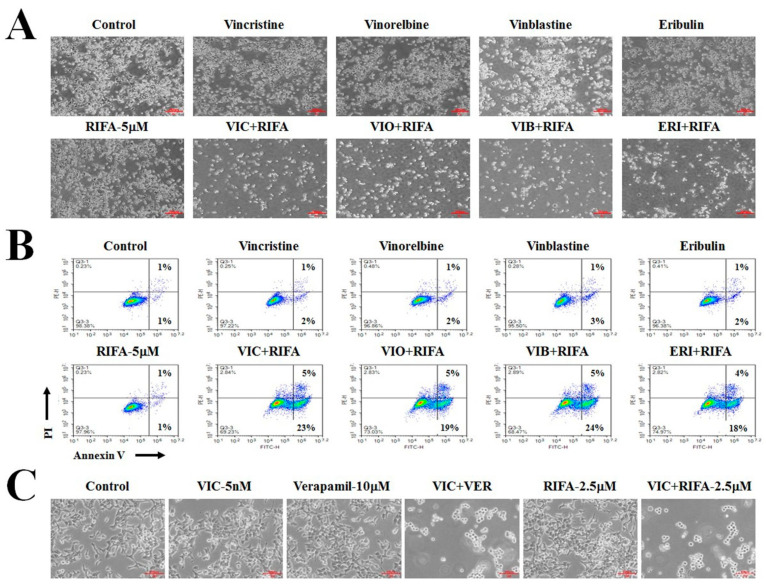
Rifabutin increases cytotoxicity of other antimitotic drugs to KBV20C cells. (**A**,**B**) KBV20C cells were treated with 5 nM vincristine, 0.1 μg/mL vinorelbine, 5 nM vinblastine, and 30 nM eribulin alone or in combination with 5 μM rifabutin (RIFA) or 0.1% DMSO (Control). After 24 h, all cells were observed using an inverted microscope at x40 magnification (**A**) or subjected to annexin V analyses (**B**). (**C**) KBV20C cells were treated with 5 nM VIC, 2.5 μM rifabutin, 10 μM verapamil, 5 nM VIC with 10 μM verapamil (VIC+VER), 5 nM VIC with 2.5 μM rifabutin (VIC+RIFA), or 0.1% DMSO (Control). After 1 day, all cells were observed using an inverted microscope at ×100 magnification.

**Figure 5 ijms-23-07383-f005:**
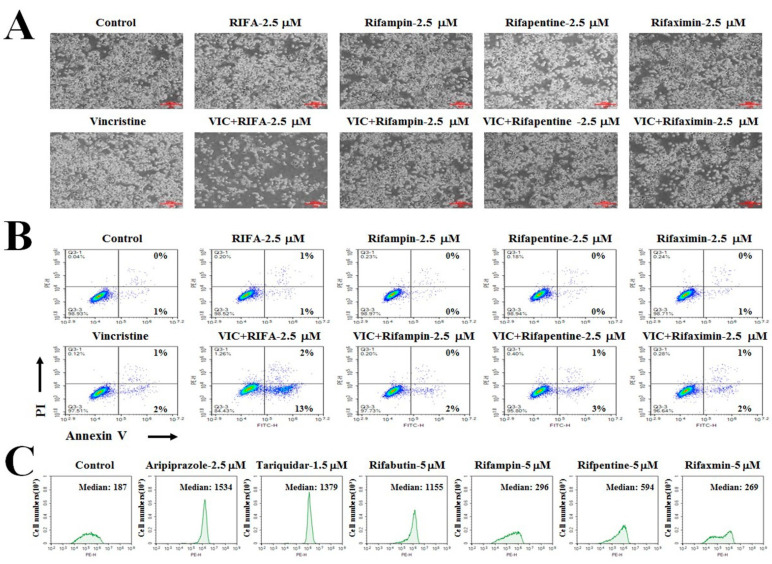
Other tuberculosis drugs (rifampin, rifapentine, and rifaximin) have little P-gp-inhibitory activity. (**A**,**B**) KBV20C cells were treated with 2.5 μM of rifabutin (RIFA), rifampin, rifapentine, and rifaximin alone or in combination with 5 nM VIC, or 0.1% DMSO (Control). After 24 h, all cells were observed using an inverted microscope at ×40 magnification (**A**) or subjected to annexin V analyses (**B**). (**C**) KBV20C cells were treated with indicated concentrations of aripiprazole, tariquidar, rifabutin (RIFA), rifampin, rifapentine, rifaximin, or 0.1% DMSO (Control). After 1 h, all cells were stained with rhodamine123 for 3 h and examined using FACS analysis.

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
