# Peer review of "Low-Dose Rifabutin Increases Cytotoxicity in Antimitotic-Drug-Treated Resistant Cancer Cells by Exhibiting Strong P-gp-Inhibitory Activity"

_ijms, 2022, doi:10.3390/ijms23137383_

Round 1

Reviewer 1 Report

Comments:

The manuscripts “Low dose of rifabutin increases cytotoxicity in antimitotic drugs-treated resistant cancer cells via strong P-gp inhibitory activity” by S.Yoon and et. al. evaluated cytotoxicity of various antibiotics at low doses to drug-resistant cancer cells and  found that rifabutin at low doses highly increased the cytotoxicity of vin-cristine (VIC) to P-gp overexpressing antimitotic drug-resistant KBV20C cells. they also tested other types of tuberculosis antibiotics (rifampin, rifapentine, and rifaximin) to de-termine whether they could increase the cytotoxicity of antimitotic drugs to P-gp overexpressing drug-resistant KBV20C cells.

In my opinion, the paper is interesting.  I believe this work will have a high impact and will be of interest for IJMS readers. Consequently, I recommend the acceptance of this work in your respected journal after revisions

I have the following comments on the manuscript

-           Revise the whole manuscript for English and typographical errors

-          Abstract is so long

-          The rational of work should be more deeply explained

-           Add conclusion of work

Author Response

The manuscripts “Low dose of rifabutin increases cytotoxicity in antimitotic drugs-treated resistant cancer cells via strong P-gp inhibitory activity” by S.Yoon and et. al. evaluated cytotoxicity of various antibiotics at low doses to drug-resistant cancer cells and  found that rifabutin at low doses highly increased the cytotoxicity of vin-cristine (VIC) to P-gp overexpressing antimitotic drug-resistant KBV20C cells. they also tested other types of tuberculosis antibiotics (rifampin, rifapentine, and rifaximin) to de-termine whether they could increase the cytotoxicity of antimitotic drugs to P-gp overexpressing drug-resistant KBV20C cells.

In my opinion, the paper is interesting.  I believe this work will have a high impact and will be of interest for IJMS readers. Consequently, I recommend the acceptance of this work in your respected journal after revisions

Response: We thank the reviewer for this comment. The manuscript has been revised according to your recommendation. Please let us know if the manuscript requires any additional modification.

I have the following comments on the manuscript

-           Revise the whole manuscript for English and typographical errors

Response: We thank the reviewer for this suggestion. Accordingly, the manuscript has been proofread by a native English speaker.

-          Abstract is so long

Response: We thank the reviewer for this comment. The Abstract has been revised and now contains less than 200 words.

-          The rational of work should be more deeply explained

Response: We thank the reviewer for this pertinent comment. The following sentences were added to the Introduction and Discussion to give a more in-depth explanation of the rationale of the study.

“Once instances of novel, FDA-approved drug repositioning are found for sensitizing P-gp-over-expressing resistant cancers, we believe that these drugs may be administered to patients with MDR cancer without the performance of further toxicity tests

Since antibiotics have been used in the clinic for a long time, if they are found to display novel mechanisms for sensitizing resistant cancers, they can be easily applied to treat cancer patients.

-           Add conclusion of work

Response: We thank the reviewer for this suggestion. A concluding statement has been added to the final paragraph of the Discussion.

Reviewer 2 Report

The manuscript #ijms-1779906 "Low dose of rifabutin increases cytotoxicity in antimitotic drugs-treated resistant cancer cells via strong P-gp inhibitory activity" by Lee et al. presentes comprehensive study in line with title. The manuscript is of a great importance and presents scientific novelty. However, I have two major issues:

First of all, all the Figures are unreadable - they should be presented in better resolution or even divided into separate Figures. Second of all, the Authors propose combination of rifabutin with vincristine (or other drugs) as competetive inhibitors of P-gp, which should help overcoming MDR phenotype of cancer/tumor cells. However, the authors have used two oral squamous carcinoma cell lines - one drug susceptible, the other MDR. The authors should also use oral normal cell line (e.g. normal epithelial) as a control to eliminate potential toxic effect of the combination to normal cells. The authors suggested that "rifabutin with strong P-gp inhibitory activity could be considered as a co-treatment to specifically target P-gp-overexpressing resistant cancer cells", which most likely is true but some exeprimental data on the above issue should be present.

Author Response

The manuscript #ijms-1779906 "Low dose of rifabutin increases cytotoxicity in antimitotic drugs-treated resistant cancer cells via strong P-gp inhibitory activity" by Lee et al. presentes comprehensive study in line with title. The manuscript is of a great importance and presents scientific novelty. However, I have two major issues:

We thank the reviewer for this comment. All revisions have been made according to your recommendations. Please let us know if the manuscript requires any additional modification.

First of all, all the Figures are unreadable - they should be presented in better resolution or even divided into separate Figures.

Response: We thank the reviewer for this comment. As Figures 1 and 2 were recognized to be content-heavy, an additional figure was included. The manuscript now has a total of five figures.

Second of all, the Authors propose combination of rifabutin with vincristine (or other drugs) as competetive inhibitors of P-gp, which should help overcoming MDR phenotype of cancer/tumor cells. However, the authors have used two oral squamous carcinoma cell lines - one drug susceptible, the other MDR. The authors should also use oral normal cell line (e.g. normal epithelial) as a control to eliminate potential toxic effect of the combination to normal cells. The authors suggested that "rifabutin with strong P-gp inhibitory activity could be considered as a co-treatment to specifically target P-gp-overexpressing resistant cancer cells", which most likely is true but some exeprimental data on the above issue should be present.

Response: We thank the reviewer for this comment. First, we are interested in sensitizing P-gp overexpressing cancer cells and not oral squamous cancer cells.

KBV20C cells were employed as a representative P-gp overexpressing cancer cell line owing to the following two reasons:

First, in our previous study (European Journal of Pharmacology 723(2014) 141147), three P-gp overexpressing resistant cancer cell lines (KBV20C, MES/SA-Dx5, and MCF7R) from different tissues (oral, uterine, and breast) were compared. However, similar results were obtained. Thereafter, only KBV20C resistant cancer cells were employed as they are easy to grow. Please see Figure 1 below.

Second, based on a previous study (Tang et al. Biochemical Pharmacology, 2014), the levels of P-gp overexpression in KBV20C cells and MCF-7/ADR were similar, with high levels of overexpression. However, the level of P-gp overexpression in these cells was less than that in P-gp transfected HEK293T cells. Naturally generated P-gp overexpressing resistant KBV20C cells are considered to be closer than P-gp transfected highly overexpressed resistant cancer cells. Please see Figure 1(A) below.

Based on previous studies by the same group, three tissue-derived P-gp overexpressing resistant cancer cells led to results similar to those obtained with oral-tissue derived resistant KBV20C cells. Please see Figure 1(B‒G) below.

As suggested, a P-gp overexpressing resistant cancer cell line from another organ, such as paclitaxcel-resistant lung cancer line, will be evaluated and applied for the development of generalized concepts to investigate or screen drugs for combination treatment in future studies. The following sentence was added to the manuscript: The results obtained with the oral KBV20C cancer cell line may be applicable to other organ-generated P-gp overexpression resistant cancer cells.”

Reviewer 3 Report

The manuscript entitled "Low dose of rifabutin increases cytotoxicity in antimitotic drugs-treated resistant cancer cells via strong P-gp inhibitory activity" (Lee et al) reported the ability of rifabutin (concentration 1 microM) to increase the antiproliferative activity of vincristine (concentration 5 nanoM) against Pgp-expressing KBV20C cells. Additionally, the authors studied the potentiating effect of other ansamycin antibiotics (namely, rifampin, rifapentine and rifaximin) highlighting that only rifabutin was able to inhibit Pgp, thus enhancing vincristine activity.

The following observations have been raised during revision:

Major comments

  • The ratio of the two agents in the effective mixture Rifabutin (1 microM)/VIC (5 nanoM) is about 500:1. This appear to be in contrast with the terms “low-dose” widely used in the manuscript by the authors. More details should be provided, also considering the active antibacterial concentration of rifabutin.
  • Rifabutin (as well as rifampin, rifapentine and rifaximin) are ansamycin antibiotics and should be named this way (and not uberculosis antibiotics or similar) in the manuscript
  • The authors should consider the difference between “dose” and “concentration” and use properly the two terms in the manuscript

Minor-comments

  • Please re-organize the abstract according to the journal guidelines (200 words maximum)
  • Please correct “canner cells” with “cancer cells”
  • Figure 1 (caption): panel E refers to the analysis after 4 h. Please correct the caption
  • For sake of clarity, please resentence “We quantitatively estimated how much each cell cycle was distributed in G1, S, and G2 stages.”
  • Paragraph 2.5: please specify the concentration of the different agents in the studied combinations
  • Not clear why verapamil has been included in the study. Please provide further details.
  • The statement “Thus, rifabutin may be used to treat P-gp-overexpressing resistant patients soon.” should be supported by animal/clinical studies. Please provide the results.
  • The conclusion paragraph is missing.
  • The abbreviation list should include only acronyms used in the manuscript
  • In the “Author Contributions” section the initials of the authors (as defined in the affiliation) should be used.
  • The authors are encouraged to use the template file that reports the lines numbers

Author Response

The manuscript entitled "Low dose of rifabutin increases cytotoxicity in antimitotic drugs-treated resistant cancer cells via strong P-gp inhibitory activity" (Lee et al) reported the ability of rifabutin (concentration 1 microM) to increase the antiproliferative activity of vincristine (concentration 5 nanoM) against Pgp-expressing KBV20C cells. Additionally, the authors studied the potentiating effect of other ansamycin antibiotics (namely, rifampin, rifapentine and rifaximin) highlighting that only rifabutin was able to inhibit Pgp, thus enhancing vincristine activity.

The following observations have been raised during revision:

Response: We thank the reviewer for this comment and appreciate the pertinent suggestions that have undoubtedly improved our manuscript. Please let us know if the manuscript requires any additional modification.

Major comments

  • The ratio of the two agents in the effective mixture Rifabutin (1 microM)/VIC (5 nanoM) is about 500:1. This appear to be in contrast with the terms “low-dose” widely used in the manuscript by the authors. More details should be provided, also considering the active antibacterial concentration of rifabutin.

Response: We thank the reviewer for this comment. First, only low-dose treatments with rifabutin were mentioned in the manuscript. As it seems the VIC treatments are viewed as low dose treatments, the entire manuscript has been revised to prevent any further confusion.

According to a prior study (BBRC, 2016), the cytotoxicity of rifabutin ranges from 5‒10 µM in lung cancer cell lines. In the present study, rifabutin was found to exhibit cytotoxic effects when a single treatment greater than 10 µM was administered to both sensitive KB and resistant KBV20C cells.

Herein, only doses less than 2.5 µM rifabutin were employed as low doses for the co-treatments. As these low doses only P-gp inhibitory effects, low doses of rifabutin were assumed to have minor toxic effects on normal cells.

  • Rifabutin (as well as rifampin, rifapentine and rifaximin) are ansamycin antibiotics and should be named this way (and not uberculosis antibioticsor similar) in the manuscript

Response: We thank the reviewer for this comment. As rifabutin, rifampin, rifapentine, and rifaximin are ansamycin antibiotics, the necessary revisions have been made throughout the entire manuscript, including the removal of parts where these drugs are described as tuberculosis antibiotics.

  • The authors should consider the difference between “dose” and “concentration” and use properly the two terms in the manuscript

Response: We thank the reviewer for this suggestion. The necessary revisions have been made in the Results section.

Minor-comments

  • Please re-organize the abstract according to the journal guidelines (200 words maximum)

We thank the reviewer for this comment. The Abstract has been revised and now contains less than 200 words.

  • Please correct “canner cells” with “cancer cells”

Response: We thank the reviewer for this suggestion. The following correction was made in the Results section: “drug-resistant KBV20C cancer cells.

  • Figure 1 (caption): panel E refers to the analysis after 4 h. Please correct the caption

Response: We thank the reviewer for this suggestion. The following revision was included: After 4 h (E), 24 h (F), or 4 h (G) of rhodamine123 staining, all cells were examined using FACS analysis.”

  • For sake of clarity, please resentence “We quantitatively estimated how much each cell cycle was distributed in G1, S, and G2 stages.”

Response: We thank the reviewer for this suggestion. The sentence was revised as follows:The number of cells distributed in the G1, S, and G2 stages following drug treatment was quantified.”

  • Paragraph 2.5: please specify the concentration of the different agents in the studied combinations

Response: We thank the reviewer for this suggestion. The sentence has been revised as follows: Cytotoxic effects of 5 μM rifabutin in combination with 0.1 μg/mL vinorelbine, 5 nM vinblastine, and 30 nM eribulin, which are antimitotic drugs used as chemotherapeutic agents in cancer patients.”

  • Not clear why verapamil has been included in the study. Please provide further details.

Response: We thank the reviewer for this comment. We recognized that the comparison between VIC+verapmail and VIC+refabutin was not important in this study.

Verapamil is a well-known P-gp inhibitor that exerts P-gp inhibitory activity in the nanomolar range based on in vitro test-tube assays. However, in a rhodamine assay with P-gp overexpressing resistant cancer cells, verapamil displayed P-gp inhibitory activity in the micromolar range.

The following sentence was added to the Results section: The sensitizing effects of a well-known P-gp inhibitor, verapamil, on VIC-treated KBV20C cells were compared to those of rifabutin.

The following sentences were also added to the Discussion: When verapamil and rifabutin were compared, rifabutin was assumed to induce nanomolar ranges of P-gp inhibitory activity in an in vitro test-tube assay. In future studies, in vitro assays should be used to determine whether rifabutin directly binds to P-gp and inhibits its activity.”

  • The statement “Thus, rifabutin may be used to treat P-gp-overexpressing resistant patients soon.” should be supported by animal/clinical studies. Please provide the results.

Response: We thank the reviewer for this suggestion. As this notion was viewed as an overstatement, it has been removed and the following sentence has been added in the Discussion: “As rifabutin has long been used in the clinic, this drug could be repositioned for P-gp overexpressing resistant cancer, enabling faster treatment of patients with drug-resistant cancer.”

  • The conclusion paragraph is missing.

Response: We thank the reviewer for this comment. A concluding statement has been added to the last part of the Discussion.

  • The abbreviation list should include only acronyms used in the manuscript

Response: We thank the reviewer for this suggestion. Appropriate revisions have been made where necessary.

  • In the “Author Contributions” section the initials of the authors (as defined in the affiliation) should be used.

Response: We thank the reviewer for this comment. The appropriate revisions have been made.

  • The authors are encouraged to use the template file that reports the lines numbers

Response: We thank the reviewer for this comment. The template file was indeed used; however, the line numbers were not included. The revised manuscript contains line numbers.

Round 2

Reviewer 2 Report

I accept Authors response. However, I still believe that the combination should be investigated against non-tumor/cancer cell line.

Author Response

I accept Authors response. However, I still believe that the combination should be investigated against non-tumor/cancer cell line.

Response: We thank the reviewer for this valuable comment and appreciate the pertinent suggestions that have undoubtedly improved our future studies. If any further changes are required, please do let us know.

As suggested, we will evaluate normal cell lines (either human skin cells or mouse cells in the future. We know that single treatment with rifabutin had similar cytotoxic effects (over 10 microM) in both resistant KBV20C and sensitive KB cells, suggesting that rifabutin is not a substrate for P-gp overexpressed on the cell membranes of resistant KBV20C cells.

The 5nM vincristine concentration for co-treatments can only be applied in P-gp overexpressing resistant KBV20C cancer cells.

If we test VIC+rifabutin in normal cells, we will consider reducing tiny amounts of vincristine since the 5nM vincristine concentration can completely kill sensitive KB cells or U87MG (glioma) cells. When we tested co-treatment experiments in non-P-gp overexpressing cells (sensitive KB (oral) or U87MG (glioma) cells), we used less than 1 nM concentrations of vincristine. 

Reviewer 3 Report

The manuscript has been amended according to the provided suggestions.

The following comments have been raised during the review:

Line 27: “tuberculosis” does not appear to be a pertinent keyword

Line 117: please deleted “displayed”

Author Response

The manuscript has been amended according to the provided suggestions.

The following comments have been raised during the review:

Line 27: “tuberculosis” does not appear to be a pertinent keyword

Line 117: please deleted “displayed”

Response: We have revised the manuscript in accordance with your recommendations.

We look forward to working with you to move this manuscript closer to publication. If any further changes are required, please do let us know.